# Fast Screening of Biomembrane-Permeable Compounds in Herbal Medicines Using Bubble-Generating Magnetic Liposomes Coupled with LC–MS

**DOI:** 10.3390/molecules26061742

**Published:** 2021-03-20

**Authors:** Xiaoting Gu, Dongwu Wang, Xin Wang, Youping Liu, Xin Di

**Affiliations:** Laboratory of Drug Metabolism and Pharmacokinetics, Shenyang Pharmaceutical University, 103 Wenhua Road, Shenyang 110016, China; guxiaoting320@126.com (X.G.); sydww2021@163.com (D.W.); wangxin68k@163.com (X.W.); yp-liu@163.com (Y.L.)

**Keywords:** bubble-generating magnetic liposomes, bionic membrane, permeable compounds, herbal medicines, LC–MS

## Abstract

A novel strategy based on the use of bionic membrane camouflaged magnetic particles and LC–MS was developed to quickly screen the biomembrane-permeable compounds in herbal medicines. The bionic membrane was constructed by bubble-generating magnetic liposomes loaded with NH_4_HCO_3_ (BMLs). The lipid bilayer structure of the liposomes enabled BMLs to capture biomembrane-permeable compounds from a herbal extract. The BMLs carrying the compounds were then separated from the extract by a magnetic field. Upon heat treatment, NH_4_HCO_3_ rapidly decomposed to form CO_2_ bubbles within the liposomal bilayer, and the captured compounds were released from BMLs and analyzed by LC–MS. Jinlingzi San (JLZS), which contains various natural ingredients, was chosen to assess the feasibility of the proposed method. As a result, nine potential permeable compounds captured by BMLs were identified for the first time. Moreover, an in vivo animal study found that most of the compounds screened out by the proposed method were absorbed into the blood. The study provides a powerful tool for rapid and simultaneous prediction of multiple biomembrane-permeable components.

## 1. Introduction

Chinese herbal medicines have been deemed as a momentous combinatorial chemical library of therapeutic agents [1]. Screening out bioactive compounds from complex herbal medicines is an important way to find new lead compounds in early drug discovery. Until now, the routine procedure of screening bioactive compounds from herbal medicines is to extract and isolate complex mixtures and then to evaluate the pharmacological activities of the individual compounds in vivo or in vitro [2]. However, this screening method is expensive and time consuming, with no consideration of the synergistic action of the multiple components in herbal medicines. Developing a more suitable method for rapid screening of active ingredients is helpful for the application of herbal medicines and the discovery of lead compounds. This study presents a novel screening strategy based on bubble-generating magnetic liposomes (BMLs) to screen the potential biomembrane-permeable components from herbal medicines.

The permeability of compounds across the cell membrane is an important prerequisite for screening the compounds that could exert their effects in the body. Liposomes are artificially prepared vesicles made of a phospholipid bilayer which are commonly used as oral delivery carriers for drugs and nutrients to increase absorption [3,4]. Due to their structural similarity to cell membranes, liposomes can mimic the cell membrane and have been used to predict the permeability of the compounds into cells. Recently, liposomes, as the simplest bionic membrane, have seen increasing use in screening biomembrane-permeable compounds or indicating the permeability of compounds through cell membranes. For instance, immobilized liposome chromatography [5,6] and liposome electrokinetic chromatography [7,8,9] are deemed to be promising approaches to investigate drug–membrane interactions, and have been employed in screening biomembrane-permeable compounds in herbal medicines. However, the brief lifespan and cumbersome preparation of the immobilized bionic membranes restrict their application. In the last few years, the strategy of screening membrane-permeable compounds in herbal medicines with liposomes by equilibrium dialysis combined with offline HPLC was developed, with the merits of easy operation and low cost [10,11,12]. However, the time-consuming equilibrium dialysis of the strategy has much room for improvement through burgeoning separation technologies such as magnetic nanoparticles (MNPs).

Magnetic separation technology has shown the benefits of avoiding the time-consuming steps of centrifugation, precipitation, and filtration [13]. MNPs have been widely used to separate various biomolecules or natural compounds from samples through a magnetic field [14,15]. The integration of MNPs and liposomes provides a rapid screening platform for biomembrane-permeable compounds [16,17,18]. In recent years, some creative strategies for controlled drug release from liposomes have been developed. Notably, bubble-generating liposomes, originally proposed by Sung [19], are produced by encapsulating NH_4_HCO_3_ in the aqueous cores of liposomes and used for controlled drug release in drug delivery. NH_4_HCO_3_ can produce CO_2_ bubbles upon heat treatment, which bestows liposomes with good temperature-triggered release properties [20]. By encapsulating NH_4_HCO_3_ and MNPs into liposomes directly, BMLs were prepared for targeted drug delivery [21,22]. Inspired by the above research, we developed a strategy based on BMLs that integrated features of capturing fat-soluble compounds into the lipid layer, magnetic separation from impurities by a magnetic field, and thermo-responsive controlled release of captured compounds to quickly screen the biomembrane-permeable compounds in herbal medicines.

As a case study, Jinlingzi San (JLZS), composed of Fructus Toosendan (mature fruits of *MeLia toosendan* Sieb.et Zucc) and Rhizoma Corydalis (dry tubers of *Corydalis yanhusuo* W. T. Wang), was selected as a model sample. As a classical traditional Chinese medicine prescription for promoting Qi circulation to relieve pain, JLZS has been commonly used in the treatment of gastric ulcers [23]. JLZS contains various natural ingredients that provide a good resource for therapeutic agents. The experimental process was as follows (Figure 1): amino-modified magnetic nanoparticles were synthesized first, then encapsulated into liposomes loaded with NH_4_HCO_3_ to form BMLs. After incubating BMLs with JLZS extract, the potential biomembrane-permeable components in JLZS were captured into the lipid bilayer of BMLs. Then, the BMLs carrying the components were separated from the extract by a magnetic field. Upon heat treatment, CO_2_, born by NH_4_HCO_3_, converged into bubbles to destroy the structure of the lipid bilayer, and the entrapped components were released and determined by LC–MS. Finally, an in vivo animal study was performed to verify whether the biomembrane-permeable components screened out by the proposed method could be absorbed into the blood of rats. The proposed screening strategy can provide a powerful tool to quickly and simultaneously predict multiple biomembrane-permeable components in herbal medicines.

## 2. Results and Discussion

### 2.1. Characterization of BMLs

FTIR spectra of samples are shown in Figure 2A. Amino-modified magnetic nanoparticles (AMNPs) showed a strong characteristic absorption peak at 578 cm^−1^ (asymmetric stretching vibration peak of Fe–O) [24], while transmissions around 1629 cm^−1^ and 3425 cm^−1^ could be assigned to the N–H stretching and bending vibrations of free –NH_2_, respectively [25], confirming the successful synthesis of AMNPs. The FTIR spectrum of blank liposomes displayed absorption peaks at 1090 and 1236 cm^−1^, attributed to symmetric and asymmetric –PO_2_ stretching vibrations; at 1467 cm^−1^, assigned to –CH_2_ bending vibration; and at around 2853 and 2924 cm^−1^, attributed to symmetric and asymmetric –CH_2_ stretching vibrations in the acyl chain [21], respectively. It should be noted that all the characteristic peaks of blank liposomes and AMNPs were observed in the spectrum of BMLs, which confirmed the successful encapsulation of AMNPs in BMLs.

The particle size distribution of AMNPs and BMLs tested by dynamic light scattering (DLS) was recorded, as shown in Figure 2B,C. The mean particle sizes of AMNPs and BMLs were 39.5 ± 0.8 nm (*n* = 3) and 235.9 ± 5.2 nm (*n* = 3). The PDI (polydispersity index) value is an indicator to evaluate the uniformity of particle sizes present in the suspension [26]. The results showed that the PDI values of AMNPs and BMLs were 0.179 ± 0.003 (*n* = 3) and 0.184 ± 0.002 (*n* = 3), respectively, reflecting the homogenous size distribution of AMNPs and BMLs. The zeta potential value indicates the stability of nanoparticles [26], the higher the absolute value of zeta potential, the higher the stability of nanoparticles. The average zeta potentials of AMNPs and BMLs were −16.03 ± 1.01 (*n* = 3), and −29.17 ± 1.27 (*n* = 3), respectively, indicating that AMNPs and BMLs were stable.

The magnetic hysteresis loops of AMNPs and BMLs, measured by vibrating sample magnetometer (VSM) at room temperature, are shown in Figure 2D. No magnetic hysteresis was observed in magnetization curves, suggesting that AMNPs and BMLs exhibited superparamagnetic properties. Generally, the covering of phospholipid on the surface of AMNPs would decrease the saturation magnetization [21]. As expected, the saturation magnetization value of BMLs (9 emu/g) was lower than that of AMNPs (27 emu/g). Although a downturn of magnetization value was observed after encapsulating AMNPs into liposomes, the saturation magnetization of BMLs could still be used to separate the BMLs from sample solution by a magnetic field.

The morphology of AMNPs and BMLs deduced from transmission electron microscopy (TEM) are shown in Figure 2E,F. The discrete AMNPs observed by TEM had a spherical shape with an average size below 50 nm, and were in the range of superparamagnetic particles. From the TEM images of BMLs, it can be seen that most of the BMLs were smooth and spherical or ellipsoidal in shape. An outer bright area of lipid film and an inside dark area of magnetic nanoparticles with high electron density suggest that the AMNPs were successfully inserted in liposomes.

### 2.2. Optimization of Rlelease Condition

The capability of hyperthermia-induced drug release in BMLs was investigated using tetrahydropalmatine (THP), the main bioactive compoment in JLZS that possesses high permeability, as a model drug. A unique temperature-dependent drug release curve is displayed in Figure 3A. It can be seen that the drug release increased with increasing temperature. The reason of thermo-responsive drug release was that the hyperthermic temperature led to the decomposition of NH_4_HCO_3_ and the generation of a large number of CO_2_ bubbles, triggering a rapid drug release [21]. The burst release of THP under hyperthermic conditions at 42 °C was caused by the cavitation force of exploding CO_2_ bubbles inside the liposomes. Peak areas of THP released from BMLs with different heating times were recorded, as shown in Figure 3B. An increasing trend of drug release from BMLs was observed with the increase of heating time, and a release equilibrium was achieved after heating for 4 min. By placing THP-loaded BMLs in the water bath at 42 °C for 4 min, THP could be released from BMLs satisfactorily.

### 2.3. Screening of Biomembrane-Permeable Compounds in JLZS

Until now, biomembrane-permeable compounds in JLZS had not been elaborated systematically. In this study, the chemical constituents of JLZS were initially identified by LC–MS. By comparing the MS splitting decomposition law and retention time with those acquired from the reference substances, 13 constituents were identified from JLZS: rutin (RUT), tetrahydropalmatine (THP), protopine (PRO), allocryptopine (ALL), jatrorrhizine (JAT), coptisine (COP), tetrahydrocoptisine (THC), tetrahydroberberine (THB), corydaline (COR), palmatine (PAL), berberine (BER), dehydrocorydaline (DHC), and toosendanin (TSN). The retention time and MS^2^ fragment ions of 13 constituents are listed in Table 1. Screening and identification of the potential biomembrane-permeable compounds from JLZS were conducted using as-prepared BMLs coupled with LC–MS. Typical full-scan product ion chromatograms of JLZS extract and reconstituted solution of decomposed BMLs are shown in Figure 4A, B. Compounds C2, C6, C7, C8, C9, C10, C11, C12, and C13 were screened out and identified as THP, COP, THC, THB, COR, PAL, BER, DHC, and TSN. The chemical structures and product ion mass spectra of the nine compounds in JLZS extract are shown in Figure 5. Among them, eight compounds (THP, COP, THC, THB, COR, PAL, BER, and DHC) were attributed to protoberberine alkaloids from Rhizoma Corydalis, while TSN was the representative limonoid from Fructus Toosendan. This result indicated that protoberberine alkaloids from Rhizoma Corydalis and limonoid from Fructus Toosendan could possess stronger liposolubility and would be more likely to penetrate biofilm to reach the target and exert the potential pharmacological effects.

### 2.4. In Vivo Animal Study

Biomembrane-permeable compounds in JLZS screened out by the proposed method are theoretically more likely to be absorbed into the blood through cells of the small intestine after oral administration of JLZS extract. To verify the practicability and effectiveness of the proposed method, the compounds of JLZS ingested into the blood of rats were identified by LC–MS and compared with the captured drugs screened out by the proposed method. Full-scan product ion chromatogram of a real plasma sample at 1 h after JLZS administration is shown in Figure 4C. It can be seen that seven components (THP, THC, THB, COR, PAL, DHC, and TSN) were absorbed into the blood of rats, while other compounds with weaker membrane permeability were not absorbed. This result suggests that the proposed strategy is effective and workable for the screening of biomembrane-permeable compound in herbal medicines or functional foods.

## 3. Materials and Methods

### 3.1. Chemicals and Materials

Egg yolk lecithin was provided by Lipoid Gmbh (Ludwigshafen, Germany). Cholesterol was purchased from Beijing Bailingwei Technology Co., Ltd. (Beijing, China). Ethylenediamine and ammonium bicarbonate (NH_4_HCO_3_) were purchased from Damao chemical reagent factory (Tianjin, China). Anhydrous sodium acetate and FeCl_3_·6H_2_O were obtained from Tianjin Hengxing Chemical Preparation CO., Ltd. (Tianjin, China). Ethylene glycol was provided by Tianjin Fuyu Fine Chemical CO., Ltd. (Tianjin, China). Fructus Toosendan and Rhizoma Corydalis were provided by GuoDa Pharmacy (Shenyang, China). COR, TSN, and PRO standards were obtained from Chendu Pufei De Biotech CO., Ltd. (Sichuan, China). The reference standards of THB, THC, DHC, and JAT were purchased from Chendu Herbpurify CO., Ltd. (Sichuan, China). THP, ALL, COP, BER, and palmatine chloride standards were provided by Chengdu MUST Bio-tech Co., Ltd. (Sichuan, China). The reference standard of RUT was provided by the National Institute for the Control of Pharmaceutical and Biological Products (Beijing, China). HPLC-grade acetonitrile was provided by Concord Technology CO., Ltd. (Tianjin, China). Formic acid (HPLC grade) was purchased from DIKMA Technologies, Inc. (Beijing, China). Deionized water used throughout the study was provided by Wahaha Corporation (Hangzhou, China).

### 3.2. Sample Preparation

The reference standards of THP, COR, THC, THB, COP, JAT, DHC, BER, PAL, PRO, ALL, RUT, and TSN were weighed accurately and dissolved in methanol/water (50:50, *v/v*) to prepare individual standard solutions with appropriate concentrations.

The smashed powder of JLZS (Fructus Toosendan 20 g + Rhizoma Corydalis 20 g) was evenly blended, then refluxed twice with 200 mL 50% ethanol for 2 h. The filtrates were pooled and condensed by rotary evaporation and then vacuum-dried. The residue was dissolved into 40 mL PBS solution and centrifuged at 4000 r/min for 30 min to obtain the supernatant JLZS extract.

### 3.3. Synthesis of Amino-Modified Magnetic Nanoparticles

AMNPs were synthesized according to the literature with slight modifications [25]. Briefly, 0.5 g of FeCl_3_·6H_2_O, 1.0 g of anhydrous sodium acetate, and 5 mL of ethylenediamine were added to 15 mL ethylene glycol and quickly stirred at 50 ± 2 °C, then transferred into the polytetrafluoroethylene autoclave when the solution was transparent. Reaction was carried out at 198 ± 2 °C for 6 h to obtain crude samples of AMNPs. Then, washing the crude samples with methanol and water twice, respectively, to remove residual solvent. Finally, the samples of AMNPs were dried at 50 ± 2 °C. The resulting AMNPs were utilized for characterization and application.

### 3.4. Preparation of Bubble-Generating Magnetic Liposomes

BMLs were prepared by the thin-film-dispersion method [21]. In brief, egg yolk lecithin and cholesterol (mass ratio 3:1) were dissolved in a 3:1 (*v/v*) mixture of dichloromethane-ethanol solution assisted with ultrasound for 5 min to obtain a phospholipid solution. Next, 10 mL of this solution was transferred to a 250 mL round-bottom flask. Then, the organic solvents were removed by rotary evaporation at 35 °C to form a uniform lipid film on the bottle wall, and vacuum-dried for 12 h to remove the residual organic solvents completely. The lipid film on the bottle wall was sufficiently hydrated by rotating with 10 mL of PBS (pH 7.0) buffer, which contained 3 M NH_4_HCO_3_ and 0.02 g dispersed AMNPs at 20 °C for 0.5 h, sonicated using a sonicator (SCIENTZ-II D) for 10 min (200 W × 4 min, 400 W × 6 min), then passed through a microfiltration membrane of 0.8 µm to obtain a liposome preliminary product. To remove the unencapsulated free AMNPs, the liposome suspension was centrifuged at 1000× *g* for 10 min. As a comparison, PBS without AMNPs and NH_4_HCO_3_ were used instead of hydration solution to prepare blank liposomes in the same way.

### 3.5. Characterization

Fourier transform infrared (FTIR) spectroscopy of the samples was performed using a Bruker IFS-55 spectrometer (Saarbrucken, Germany), with blending and further compressing of the samples with KBr to form a pellet. A vibrating sample magnetometer (VSM) (BKT-4500Z, China) was used to measure the magnetic hysteresis loops of samples with a maximum magnetic field of 20 kOe. The particle size distribution, PDI, and zeta potential of the samples were evaluated by dynamic light scattering (DLS) using a Zetasizer (Nano ZS, Malvern, UK) by suspending the sample in deionized water. The suspensions of BMLs or AMNPs were diluted with deionized water and dropped on the copper-coated grid separately, followed by staining with 2% phosphowolframic acid (*w/v*) for 30 s. Then, the samples were dried and examined under a high-resolution transmission electron microscope (TEM) (JEM2100, JEOL, Japan). Finally, the morphology of the prepared samples was deduced from the TEM.

### 3.6. LC–MS Instrumentation

The LC–MS system was a Thermo TSQ Quantum Ultra triple-quadrupole mass spectrometer equipped with an electrospray ionization source (San Jose, CA, USA). LCquan quantitation software (version 2.5.6, Thermo Fisher Scientific, Waltham, MA, USA) was used to control the LC–MS system for data acquisition and analysis. Chromatographic separation was conducted using a Waters XTerra^®^ MS C18 column (3.0 mm × 50 mm, 5 µm) maintained at 20 °C. By injecting 5 µL samples with flow rate of 0.2 mL/min, chromatographic separation was achieved using a mobile phase of 0.1% formic acid water (A) and acetonitrile (B) via gradient elution, 0–2 min, 20% B; 2–3 min, 30% B; 3–20 min, 30% B. The analytes were detected in ESI source with positive and negative ion modes in individual runs with multiple reaction monitoring.

### 3.7. Optimization of Release Condition

The properties of hyperthermia-induced bubble generation and drug release from BMLs were evaluated using THP as a model drug to optimize the temperature and time required for drug release from thermosensitive BMLs. First, the suspension of BMLs was incubated with THP solution at 4 °C for 30 min. BMLs carrying THP were separated from the impurities by a magnetic field, discarding the upper suspension and eluting the BMLs with PBS buffer 6 times; an equal volume of fresh PBS buffer was used to suspend the BML sample. Then, the hyperthermia-induced drug release of BMLs was studied by immersing the samples in a water bath at various temperatures (35, 37, 40, 42, 45 °C) for different amounts of time (1, 2, 3, 4, 5 min). The peak areas of THP in PBS buffer at specific conditions were determined by an LC–MS system to obtain the optimal temperature and time required for burst drug release from BMLs.

### 3.8. Screening of Biomembrane-Permeable Compound in JLZS

1 mL of JLZS extract was incubated with 1 mL of BMLs in a glass test tube at 4 °C for 30 min, with occasional gentle shaking for ample reaction. BMLs carrying biomembrane-permeable compounds were separated from the suspension by magnetic separation; the upper suspension was discarded and the BMLs were further eluted with PBS 6 times. Then, the BMLs were placed in a water bath at 42 °C for 4 min to release the drugs. After separating the released drugs by magnetic attraction, the top layer was collected and centrifuged at 12,000 × *g* for 4 min. Finally, 5 µL of the as-obtained supernatant was used to screen the permeable compounds in JLZS by LC–MS.

### 3.9. In Vivo Animal Study

Six healthy male Sprague Dawley rats (about 220 g) were provided by Liaoning Changsheng Biotechnology Co., Ltd. (SYXK2018-0009, Shenyang, China) and kept in a controlled environment with a 12 h/12 h light/dark cycle. The animal experiment program was approved by the Animal Ethics Committee of Shenyang Pharmaceutical University (SYPU-IACUC-C2020-12-11-101). After 12 h of fasting, but with free access to water, rats received gavage administration of 1.35 g/kg for JLZS extract (calculated as raw herbs). At 15 min, 30 min, 1 h, and 2 h after oral administration, 0.3 mL blood samples were collected from the retro-orbital vein of each rat, placed in heparinized tubes, and further centrifuged immediately at 3500× *g* for 10 min at 4 °C to separate the plasma. Plasma samples were pretreated by precipitation protein with methanol. In brief, we mixed 50 µL plasma with 150 µL methanol, vortexed it for 1 min, and further centrifuged it at 12,000× *g* for 4 min. Finally, 5 µL of supernatant was injected into the LC–MS system.

## 4. Conclusions

In this study, BMLs were prepared and characterized successfully, and a screening strategy using BMLs coupled with LC–MS was developed for the first time and successfully used to quickly screen the biomembrane-permeable compounds from JLZS. As a result, nine compounds (THP, COP, THC, THB, COR, PAL, BER, DHC, and TSN) of JLZS were screened out which possessed stronger liposolubility and would be more likely to penetrate biofilms to reach their target and exert potential pharmacological effects. The as-prepared BMLs are promising for the rapid screening of biomembrane-permeable compounds from herbal medicines in vitro. 

## Figures and Tables

**Figure 1 molecules-26-01742-f001:**
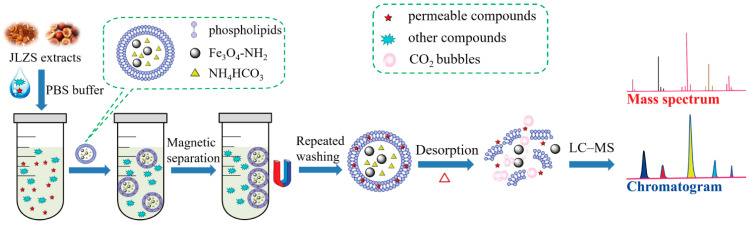
Schematic illustration of the proposed screening strategy using bubble-generating magnetic liposomes coupled with LC–MS.

**Figure 2 molecules-26-01742-f002:**
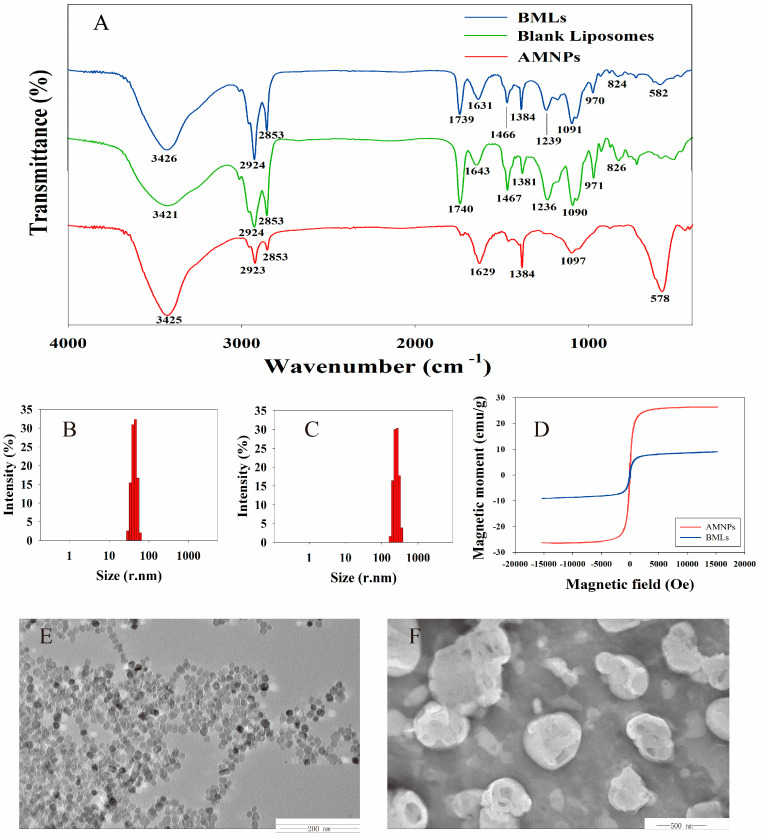
Characterization of bubble-generating magnetic liposomes (BMLs): FTIR spectroscopy of BMLs, blank liposomes, and amino-modified magnetic nanoparticles (AMNPs) (**A**). Particle size distribution of AMNPs (**B**) and BMLs (**C**). Magnetic hysteresis loops of BMLs and AMNPs (**D**). TEM images of AMNPs (**E**) and BMLs (**F**).

**Figure 3 molecules-26-01742-f003:**
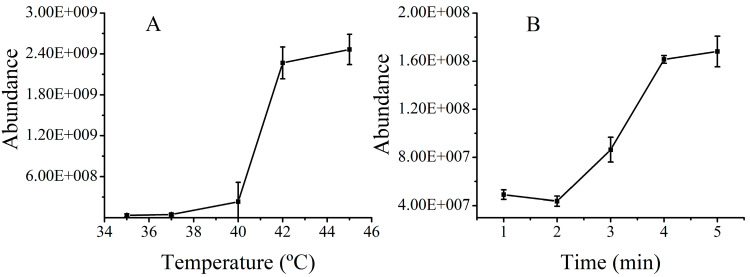
Optimization of the temperature and heating time for the destruction of liposomes: Peak areas of tetrahydropalmatine (THP) released from BMLs at different temperatures for 4 min (**A**). Peak areas of THP released from BMLs at 42 °C with different heating times (**B**).

**Figure 4 molecules-26-01742-f004:**
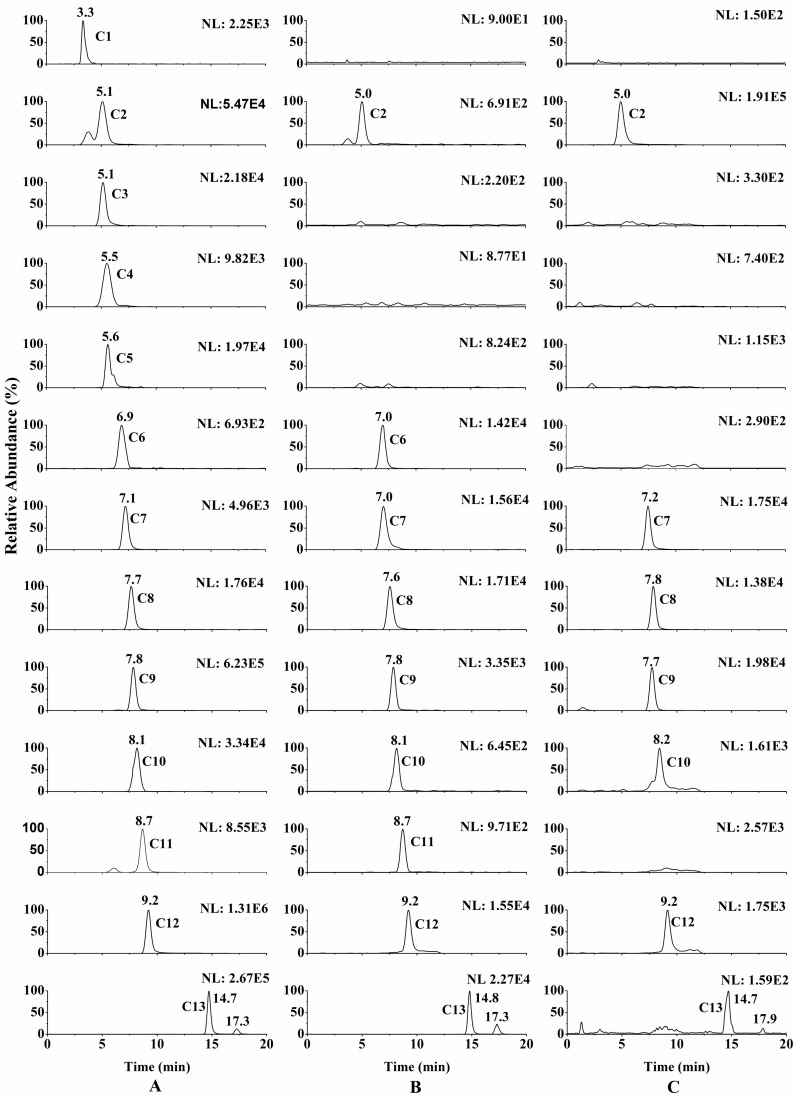
Typical full-scan product ion chromatograms of Jinlingzi San (JLZS) extract (**A**), reconstituted solution of decomposed BMLs (**B**), and real plasma sample at 1 h after JLZS administration (**C**).

**Figure 5 molecules-26-01742-f005:**
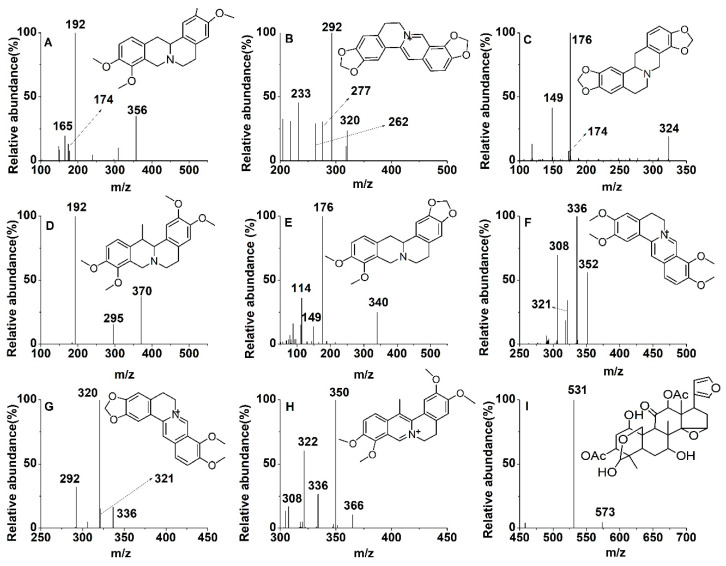
Product ion (MS^2^) spectra and chemical structures of tetrahydropalmatine (THP) (**A**), coptisine (COP) (**B**), tetrahydrocoptisine (THC) (**C**), corydaline (COR) (**D**), tetrahydroberberine (THB) (**E**), palmatine (PAL) (**F**), berberine (BER) (**G**), dehydrocorydaline (DHC) (**H**), and toosendanin (TSN) (**I**).

**Table 1 molecules-26-01742-t001:** Retention time (t_R_) and MS^2^ fragment ions of thirteen components identified in JLZS.

Peak No.	Compound	Formula	t_R_ (min)	MS	MS^2^
C1	RUT	C_27_H_30_O_16_	3.3	609	300, 271, 178
C2	THP	C_21_H_25_NO_4_	5.1	356	192, 174, 165
C3	PRO	C_20_H_20_NO_5_	5.1	354	336, 275, 189,188
C4	ALL	C_21_H_24_NO_5_	5.5	370	352, 290, 206, 188
C5	JAT	C_20_H_20_NO_4_	5.6	338	321, 303, 237
C6	COP	C_19_H_14_NO_4_	6.9	320	292, 277, 262, 233
C7	THC	C_19_H_17_NO_4_	7.1	324	176, 174, 149
C8	THB	C_20_H_21_NO_4_	7.7	340	176, 149, 114
C9	COR	C_22_H_27_NO_4_	7.8	370	295, 192
C10	PAL	C_21_H_22_NO_4_	8.1	352	336, 321,308
C11	BER	C_20_H_18_NO_4_	8.7	336	321, 320, 292
C12	DHC	C_22_H_24_NO_4_	9.2	366	350, 336, 322, 308
C13	TSN	C_30_H_38_O_11_	14.7, 17.3	573	531

## Data Availability

Data are contained within the article.

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
