# Peer review of "Fast Screening of Biomembrane-Permeable Compounds in Herbal Medicines Using Bubble-Generating Magnetic Liposomes Coupled with LC–MS"

_molecules, 2021, doi:10.3390/molecules26061742_

Round 1

Reviewer 1 Report

In this submitted manuscript (molecules-1110075), new drug discovery platform using BML for identification of ingredient from Chinese herb medicine (JLZS) was developed and its feasibility (practicability and effectiveness) was verified. This method is included some unique ideas, such as hyperthermia liposome and magnetic separation. However, I have several minor concerns.

I could not understand the advantage of liposome usage for screening of herb ingredients. I thought that LC-MS analysis of the extract with organic solvent from herb extract is easier procedure than this platform. Please explain the merits and usefulness of liposome usage.

The particle size of BML was shown in Figure 2B. Please measure that of AMNP in order to compare both.  

In Figure 3, optimal condition for releasing from BLM, was provided. Please show the time (Figure 2A) and temperature (Figure 2B).

In Figure 4 (a) and (b), LC-MS charts were presented. But, It isn't possible to compare an ingredient of these 13 quantitatively in individual chart. I would like to know all Integrate chart or table indicated quantitative values.

Was the compound released from BML only just 9 ingredients? Was not another compound detected, except of 13 authentic compounds, such as glaucine from Rhizoma Corydalis and 21-O-acetyltoosendantriol from Fructus Toosendan?

In the method, ingredients from JNZS in rat plasm were detected at four independent time after oral administration of JLZS extract. But, the result of only 1 h was presented. I would like to know time course of each compounds.

The accession No. of the animal experiment provided from your animal ethics committee should be described.

I’m pleasure that these comments will be helpful for you.

Reviewer 2 Report

This paper developed a novel research method on screening of the bioactive compounds from the complex mixtures of herbal medicines, which was based on bubble-generating magnetic liposomes to discover the potential membrane-permeable components from herbal medicines. Finally nine potential permeable compounds captured by the lipid bilayer of bubble-generating magnetic liposomes were authenticated by LC-MS for the first time. In a word, this research provided a way to quick predict multiple membrane-permeable components. Here are some issues need to be addressed.

  1. In page 3 “Characterization of BMLs” section “the average zeta potential tested by DLS …… AMNPs were encapsulated into liposomes successfully to form BMLs.” As for liposomes, it’s easy to aggregate due to its natural charge, which could be seen in Figure 2D, but the AMNPs with a negative value (-4.35 mV) and the neutral charge of egg yolk lecithin, why were the BMLs a little positive value (0.13 mV). Please provide the zeta potential of blank liposomes (egg yolk lecithin).
  2. The BMLs showed multilamellar spheres, the polydispersity index values (0.254) was slightly higher. The description diagram of particle size was generally recommended to use the Diameter-Intensity diagram. The permeable compounds needed much time to achieve the equilibrium for the multilamellar liposomes and the saturation magnetization value of BMLs (9 emu/g) was lower than that of AMNPs (27 emu/g), whether it affected the partition of weakly permeable compounds in BMLs? What were the main factors for the partition of these compounds in BMLs? Especially, the AMNPs were unevenly encapsulated in BMLs from the Figure 2D.
  3. Whether the experimental method proposed in this study was limited by the strong or weak permeability of the compound extracted from the complex mixtures of herbal medicines, because other compounds with weaker membrane permeability were not absorbed. If so, this approach might be suitable for potential active drugs with stronger liposolubility.
  4. There are a few nonstandard formats, pay attention to add a space between numbers and characters (such as “palmatine(PAL)” in 2.3 section, “3M NH4HCO3” in 3.4 section), and the expression should be consistent (“Figure 2A”, “Figure 2B”, “Figure 3 (A)” in page 4 and 5). Please check the whole manuscript again.
  5. The English language needs to be improved. For example, “herb medicines” can be changed to “herbal medicines”. The word “novel” can be deleted.

Round 2

Reviewer 2 Report

Most of the comments have been well-addressed, and the manuscript has been improved. There are some minor issues need to be further addressed.

1, The third paragraph can be further revised. The authors discussed about the liposome was from the drug discovery aspect, but this study was the discovery of membrane permeable compounds from traditional Chinese medicine. “The demulsifiers are harmful to the human body and the environment.” may not be the main disadvantage of those previous reported method.

2, Section “2.3. Optimization of LC–MS Conditions”, which only provides little useful information, can be deleted. Or only described by a few sentences in the section 2.4.

3, Please provide a total ion chromatograms of the sample, at least three of the total ion chromatograms can be added. The samples before and after treatment by liposomes, as well as the liposomes released sample.

4, Please further check about the explanation. For example, “authentication” is not proper for compound, the” identification” may be better.
